

# Soil erosion evolution and spatial correlation analysis in a typical karst geomorphology, using RUSLE with GIS

**Cheng Zeng**[1,2,3], **Shijie Wang**[1,3], **Xiaoyong Bai**[1,3], **Yangbing Li**[2], **Yichao Tian**[1,3], **Yue Li**[4], **Luhua Wu**[1,3] , **Guangjie Luo**[3,5]

[1] State Key Laboratory of Environmental Geochemistry, Institute of Geochemistry, Chinese Academy of Sciences, 99 Lincheng West Road, Guiyang 550081, Guizhou Province, PR China

[2] School of Geography and Environmental Sciences, Guizhou Normal University, Guiyang 550001, China

[3] Puding Karst Ecosystem Observation and Research Station, Chinese Academy of Sciences, Puding 562100, Guizhou Province, PR China

[4] Beijing Forestry University; Key Laboratory of State Forestry Administration on Soil and Water Conservation, Beijing, 100083, China

[5] Institute of Agricultural Ecology and Rural development, Guizhou Normal College, Guiyang 550018, China

*Correspondence to:* Xiaoyong Bai (baixiaoyong@126.com)

**Abstract.** In spite of previous studies on soil erosion in Karst landform, limited data are available regarding the spatial and temporal evolution and the correlation of spatial elements of soil erosion in Karst. The lack of this study leads to misassessment of environmental effects on the region especially in the mountainous area of Wuling in China. Soil erosion and rocky desertification in this area influence the survival and development of 0.22 billion people. For this reason, the typical Karst area in South China is the object of this study. This paper aims to analyze the spatial and temporal evolution characteristics of soil erosion and investigate the relationship between soil erosion and rocky desertification by using GIS technology and modified universal soil loss equation (RUSLE) model to reveal the relationship between soil erosion and major natural elements in this area. (1) In 2000–2013, the proportion of the area of micro- and slight erosion increases, whereas the proportion of the area of moderate erosion and above decreases. Erosion of moderate and above levels changes into micro- and slight erosion. (2)The soil erosion area in slope zones at 15°–35° accounts for



60.59% of the total erosion area and 40.44% of total erosion. (3) The amplitude reduction in the annual erosion rate is higher in the Karst area than that in the non-Karst area. Soil erosion in different outcrop areas of rock generally shows an improving trend, but the dynamic changes in soil erosion significantly differ among various lithological distribution belts. (4) The soil erosion rate of rocky desertification area with moderate and below levels of erosion decreases, whereas the erosion rate of rocky desertification area with severe erosion level increases. Results show the gradual decrease in the temporal and spatial variation of soil erosion in the study area. Lithology is the geological basis of soil erosion. Changes in the spatial distribution of lithology and rocky desertification induce high soil loss. The area is characterized by high rocky desertification, low erosion module, and decreasing annual erosion rate.

## 1  Introduction

Soil erosion is one of the most serious environmental problems that affect global ecological environment and human development(Higgitt, 1991; Martínez-Casasnovas, 2016; Borrelli, 2016). This phenomenon causes the loss of soil nutrients and land degradation and exacerbates the occurrence of drought, floods, landslides, and other disasters(Munodawafa, 2007; Park et al., 2011; Rickson, 2014; Arnhold et al., 2014);serious soil erosion directly influences the development, application, and protection of regional resources(Cai and Liu, 2003; Ligonja, 2015). Soil erosion threatens the regional and even global ecological security patterns.

The evolution of soil erosion in Karst area is often related to many factors(Karamesouti, 2016; Krklec et al., 2016; Wang et al., 2016; Wu L et al., 2016,)because of its complicated natural conditions(Bai et al., 2013a; Bai et al., 2013b; Tian et al,, 2016). Therefore, the spatial evolution of soil erosion in Karst area and its influencing factors must be evaluated for the development of research on the ecology and soil erosion in the area. In the context of global soil erosion and land degradation, traditional studies on runoff plot and watershed hydrologic station cannot maximize the use of soil erosion data in Karst. Hence, the basic research on soil erosion in Karst area is the basis of water and soil conservation.



China possesses the most concentrated, widely distributed, and most complicated Karst areas.
Guizhou province is the center and typical representative of the south Karst areas in China. Soil
erosion in the Karst area exhibits slow soil formation rate, mismatched water and soil space,
particular geological and hydrological background and underground structure(Wang and Li, 2007);
as such, determining soil erosion in the Karst area is more complex and special than that in
non-Karst area. Soil erosion in the Karst area is related to topography, lithology, and rocky
desertification. In addition to the surface loss, underground leakage is observed in the area. The
Karst area has small environmental capacity and low restorability of the ecological
system(Wallbrink, 2002). Soil erosion has serious consequences and can restrict the sustainable
development of the regional social economy.
Many scholars studied soil erosion and determined the cause of soil erosion and the
characteristics of its spatial evolution. Erosion force(Bai and Wan, 1998; Feng et al., 2011), erosion
process(Edgigton et al., 1991; Cao et al., 2012),soil degradation(Feng et al., 2016; Gao et al., 2015;
Guo et al., 2015), and erosion mechanism(Hancock et al., 2014)have also been explored. Currently,
studies on soil erosion are mainly concentrated in non-Karst areas or international basins(Fernández
and Vega, 2016; Park et al., 2011;). Limited studies investigated the fragile ecological geological
environment within the Karst area. Some scholars also conducted preliminary studies on soil erosion
in the Karst landform area. For example, Li et al. (2016)calculated soil erosion in a typical Karst
basin by using the RUSLE model and discussed the influence of slope on the temporal and spatial
evolution laws of soil erosion in the Karst area; the result shows that the area within the slope of
8°–25° is the main erosion slope in the basin.Yang et al. (2014) estimated soil erosion in
Chaotiangong County, Guilin by using analytic hierarchy and fuzzy model; the result shows that the
risk of soil erosion is very high in southeast of the study area and is relatively low in the northwest
area. Biswas and Pani(2015) studied soil erosion of Barakar basin in East India by using the RUSLE
model combined with GIS technology; soil erosion of more than 100 t/(hm$^2$·a) accounts for only
0.08% of the total study area.Feng et al. (2016)compared the soil erosion rates of two Karst peak
cluster depression basins in northwest of Guangxi, China by using $^{137}$Cs and RUSLE model; runoff



discontinuity and underground seepage in Karst slope are significant because they effectively reduce
the effect of the slope length in the RUSLE model. However, some deficiencies and defects were
found in the previous studies. For the selection of research areas, the most selected the Karst basins
or mountains to make study; as analyzing driving factors, most studies analyzed the effect of terrain,
rainfall, vegetation cover, and other factors on soil erosion. The response of rocky desertification and
lithology to soil erosion is ignored. Few scholars analyzed the soil erosion evolution in Karst valley
area in the long time sequence, and few scholars use the effect of spatial factor on soil erosion
evolution in Karst. Therefore, data on the correlation analysis on soil erosion evolution and spatial
factors in the Karst area is rare, especially in the mountainous area of Wuling, China. The lack of this
study leads to a miscarriage of justice in the assessment of environmental effects in the region. Soil
erosion and rocky desertification in this area influence the survival and development of 0.22 billion
people. Studying the temporal and spatial distribution evolution of soil erosion in the Karst area and
its correlation to spatial factors by using effective means and method remains a problem. Research
on this aspect is internationally scarce and rare; support on data, as well as experience and
contribution of technical methods are lacking.

This paper focuses on typical Karst areas in South China and analyzes the soil erosion in

different periods by using the modified universal soil loss equation (RUSLE) combined with an
actual survey on soil types and the calculation results of soil corrosion test to solve the following
problems: (1) identify the temporal and spatial distribution evolution of soil erosion in typical Karst
areas in the south of China; (2) identify the relationship between soil erosion and rocky
desertification; (3) reveal the correlation between soil erosion and master natural elements, and
evaluate its ecological effect. Raise the improvement and suggestions on research method and
research emphasis. This study provides the basis for the macro decision-making of government
policy makers and environmental managers, as well as the experience in methodology and reference
in the data for international counterparts to study the soil erosion in Karst landform area.

**2   Study area**




Yinjiang County is located in northeast Guizhou plateau(China), Yinjiang rivers of Wujiang River
water system in the Yangtze River basin watershed areas(Fig.1(a)(b)). The geographical position of
the study area is 108°17' to 108°48'N, 27° 35' to 28°28'E, and the land areas is 196900 hm². Fanjing
Mount, the main peak of Wuling mountains is located in the east of Yinjiang, with topography of
east high and west low, sloping from southeast to northwest, with relative elevation of 2000 m and
average elevation of 2480 m(Fig.1(d)).

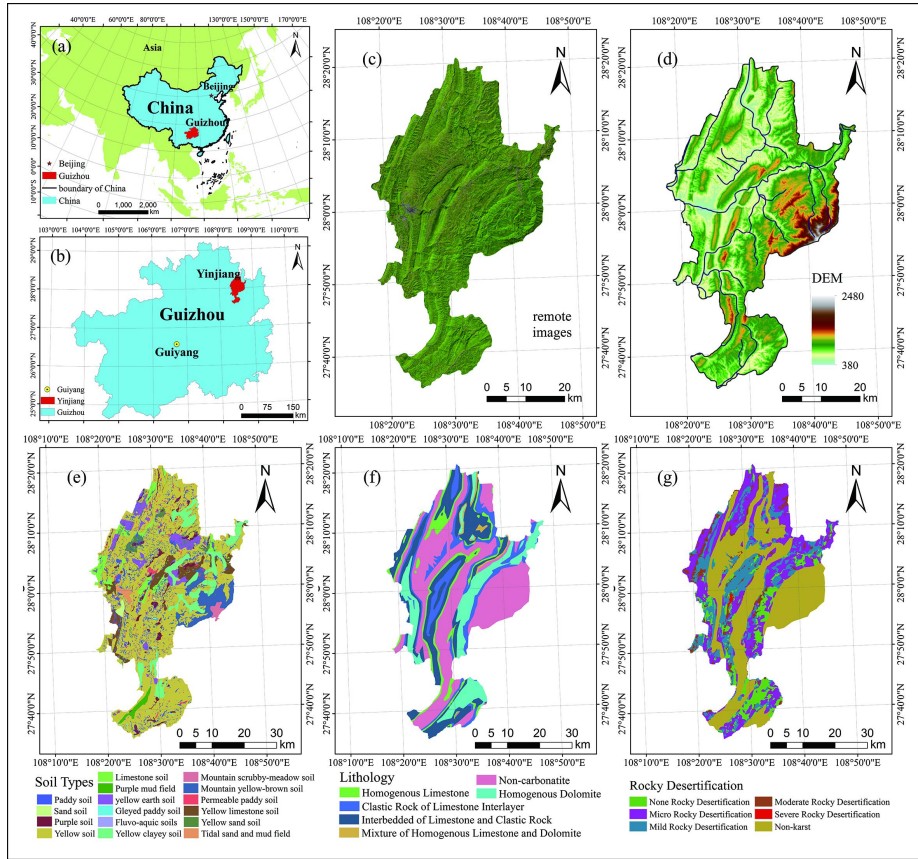


**Figure 1.** Study area location in Guizhou, China (a)(b), Study area remote images(c), Topography
(d), Soil map (e), Lithology (f) and Rocky desertification (g)

The study area has a subtropical monsoonal climate with an annual precipitation of 1100 mm.
Rainfall mainly occurs between April and August. The temperature on this area ranges from −3.1 °C



to 29.8 °C, with an annual average of 16.8 °C. The highest monthly temperature occurs in July and
the lowest occurs in January. Main vegetation includes evergreen broad-leaved forest, coniferous
forest, evergreen deciduous broad-leaved mixed forest, and temperate coniferous mixed forest. The
vegetation coverage rate increased from 49.1% to 58.5% during the study periods. Carbonate rocks
are widely distributed in Yinjing County, accounting for 60.06% of the total area. Under the action of
Karst, the mantle rock is discontinuous with underground fissure and Karst development(Fig.1(f)).
Widely distributed soil erosion led to thin soil layer in the study area, fragile ecological system.
Yinjing County suffered from different degrees of rocky desertification, accounting for 57.69% of
the total area of the whole county(Fig.1(g)). According to the classification of soil zonality, the study
area has yellow soil, but a large area is distributed with limestone. Moreover, based on the site
survey, mountain shrub meadow soil, soil mud, purple mud field, tidal sand mud field, and other soil
types are distributed in Yinjing(Fig.1(e)). All these factors are increased in a typical Karst area.

## 136    3   Materials and methods

### 137    3.1   Data Sources

The collected data in this paper included the monthly rainfall data in the study area in 2000, 2005,
and 2013 from Tongren Meteorological Bureau. The soil database was established according to the
actual survey on soil types, particle size, and content of organic substance of various soil types that
are mainly based on China soil. DEM was obtained from China remote sensing satellite ground
station, Chinese Academy of Sciences(http://www.cas.cn), with spatial resolution of 30 m. *NDVI* and
*VFC* data were from China geospatial data cloud platform(http://www.gscloud.cn). Landsat 7 OLI
and Landsat 8 OLI remote sensing images (P126, R40 and P126, R41) were synthesized in
ArcGIS10.0 for stitching and cutting, with the data from China geospatial data cloud platform, with
spatial resolution of 30 m. Based on these data, the land-use map was drawn in ArcGIS10.0 software.
Albers Conical Equal Area was used for the geographic coordinate system.



## 3.2 The RUSLE model

RUSLE model(Renard et al., 1997) is an empirical soil erosion prediction model modified from USLE model. The calculation formula is as follows:

$$A = R \cdot K \cdot L \cdot S \cdot C \cdot P \qquad (1)$$

where $A$ refers to the amount of soil loss per unit area in time and space. The unit of soil erosion depends on the units of $K$ and $R$. Many studies adopted the US unit t/(hm$^2$ ·a). $R$ refers to the rainfall erosivity factor in consideration of the erosion of snow melting runoff, in the international unit of MJ·mm/(hm$^2$ ·h·a). $K$ refers to the soil erodibility factor, which means that the soil loss rate of a certain given soil rainfall erosivity per unit is measured in a standard plot, with the international unit of t·hm$^2$ ·h/( hm$^2$ ·MJ·mm). $LS$ refers to the slope aspect factor. $C$ refers to the coverage factor of vegetation. $P$ refers to the conservation measure factor, including engineering measure and tillage measure factor.

### 3.2.1 Rainfall erosivity factor($R$)

Rainfall erosivity is the potential ability of rainfall induced erosion. Rainfall erosivity is the primary factor in soil loss equation and is related to rainfall, duration of rainfall, and rainfall energy. This factor reflects the effect of rainfall characteristics on soil erosion. Directly measuring the rainfall erosivity is difficult. Most studies adopt the rainfall parameters, including rainfall intensity and precipitation rain fall to estimate the rainfall erosivity. Given the relatively fragmented surface, concentrated precipitation, and strong water erosion in the study area, this paper adopts the simple formula of monthly rainfall by Zhou Fujian et al.(1995) to estimate the rainfall erosivity ($R$) in Yinjiang by comparing various algorithms and the accuracy of acquired climate data. The formula is as follows:

$$R = \sum_{i=1}^{12}(-1.5527 + 0.7297P_i) \qquad (2)$$

where $P_i$ refers to the rainfall in month $i$ (mm). The unit of calculated $R$ is 100ft·t·in·ac$^{-1}$·h$^{-1}$·a$^{-1}$. If $R$ is changed to the international unit MJ·mm·hm$^{-2}$·h$^{-1}$·a$^{-1}$, then the coefficient 17.02 should be multiplied (Table 1).





**Table 1.** The rainfall erosivity factor ($R$) in Yinjiang during the study periods

| Year | Annual rainfall (mm) | The annual rainfall erosivity [MJ·mm·hm⁻²·h⁻¹·a⁻¹] |
|---|---|---|
| 2000 | 1121.03 | 3183.25 |
| 2005 | 884.23 | 2460.92 |
| 2013 | 734.39 | 2003.93 |

### 3.2.2  Soil erodibility factor($K$)

Soil erodibility is an important indicator that reflects the rainfall infiltration capacity of soil, and the sensitivity of soil to rainfall and runoff erosion, and carry, and it is an internal factor of influencing soil loss. The size of $K$ value is related to soil texture and the content of organic material. In this paper, soil erodibility and soil mechanical composition are used to form the calculation formula with close relation to the content of organic carbon(Sharpley and Williams, 1990):

$$K = \left\{ 0.2 + 0.3 \exp\left[ -0.0256\, SAN \left( 1 - \frac{SIL}{100} \right) \right] \right\}$$
$$\times \left( \frac{SIL}{CLA - SIL} \right)^{0.3} \times \left( 1 - \frac{0.7\, SN1}{SN1 + \exp(-5.51 + 22.9\, SN1)} \right) \qquad (3)$$

where $K$ refers to the soil erodibility in the US unit ((t·acre·h)/(100·acre·ft·tanf·in)). However, the international unit is ((t·hm²·h)/(hm²·MJ·mm)); hence, a conversion factor of 0.1317 should be multiplied. *SAN, SIL, CLA*, and *C* refer to the sandy particles (0.050-2.000mm), the powder particles (0.002-0.050mm), the clay particles (<0.002mm), and the content of organic material (%); *SN1 =1-SN/100*. Different $K$ values are obtained from different soil types in the soil type map Fig.2(a).

### 3.2.3  Topographic factor($L$)($S$)

The slope length factor is a basic terrain factor that influences soil erosion. In this paper, the study result of Liu Baoyuan et al.(2000) is used to calculate the slope length in Yinjiang County:

$$S = \begin{cases} 10.8\sin\ \theta + 0.03, \quad, \quad \theta \lhd 5° \\ 16.8\sin\ \theta - 0.05, \quad,\ , \ 5° \leq \theta \lhd 10° \\ 21.9\sin\ \theta - 0.96, \quad, \quad \theta \geq 10° \end{cases} \qquad (4)$$

$$L = (\lambda / 22.13)^m \qquad (5)$$

where $S$ refers to the slope factor, θ refers to the slope value (°), $L$ refers to the slope length factor,





and λ refers to the slope length (m). First, 30m DEM data is used to extract the slope and length
from ARCGIS, and are subsequently placed in the formula to calculate the length factor $L$, slope
factor $S$, and slope length factor $LS$ as shown in Fig.2(b)(c).
**3.2.4    Vegetation cover factor($C$)**
A good correlation exists between the vegetation cover and C value; hence, this paper used $NDVI$ of
$MODIS$ as the data resource to calculate the vegetation coverage factor $C$ (formula 1) based on the
methods of Cai Congfa et al.(2000), as well as the vegetation coverage rate by referring to the
algorithm by Tan Binxiang et al.(2005)
$$C = \begin{cases} 1, \ f_c = 0 \\ 0.6508 - 0.3436 lg f_c, 0 \lhd f_c \lhd 0.783 \\ 0, f_c \geq 0.783 \end{cases} \qquad (6)$$

$$f_c = (NDVI - NDVI_{soil})/(NDVI_{veg} - NDVI_{soil}) \qquad (7)$$

$$NDVI = \rho_{NIR} - \rho_R / \rho_{NIR} + \rho_R \qquad (8)$$

where $C$ refers to the vegetation coverage factor, $f_c$ refers to the vegetation coverage (%), $NDVI$
refers to the normalized differential vegetation index, $NDVI_{veg}$ refers to the $NDVI$ value of pure
vegetation cover pixel, and $NDVI_{soil}$ refers to the $NDVI$ value of bare soil cover pixel. In this paper,
the cumulative percentages of 5% and 95% are used as confidence interval to read out the
corresponding pixel values to determine the effective $NDVI_{soil}$ and $NDVI_{veg}$ in the study area. $\rho_{NIR}$
refers to the near infrared band, and $\rho_R$ refers to the red band. The above formula is used to
calculate the vegetation coverage distribution map in different periods as shown in Fig.2 (e) (f) (g).
**3.2.5    Conservation practice factor($P$)**

**Table 2.** Soil and water conservation measure factors in Yinjiang County

| Land use types | Forest | Grassland | Cropland | Paddy field | Town | Village | Road | Waters | Unused land |
|---|---|---|---|---|---|---|---|---|---|
| p | 1 | 1 | 0.4 | 0.15 | 0 | 0 | 0 | 0 | 1 |

Soil and water conservation measure factor $P$ refers to, after adopting soil and water conservation
measure, soil loss amount, comparing with that as planting down the slope, is in the range of 0–1. If



the value is 0, it represents the area without soil erosion; if the value is 1, it represents the area
without any soil and water conservation measure (Table 2).

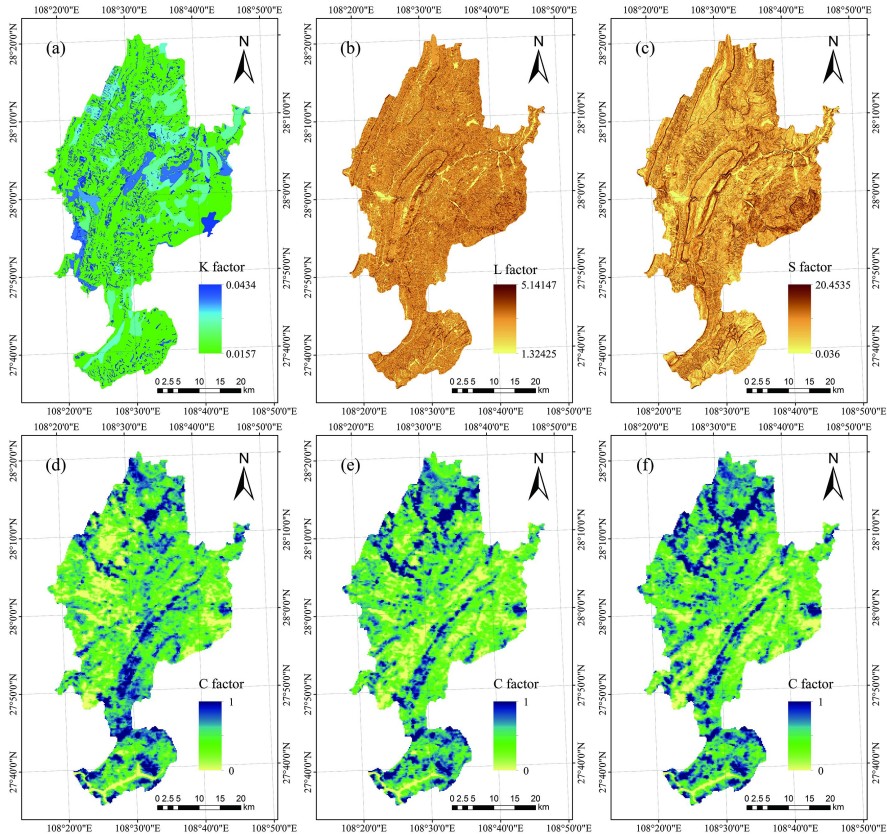


**Figure 2.** Soil Erodibility map(a), Slope Length Factor map(b), Slope Gradient Factor map(c), 2000

Vegetation Cover Factor map(d), 2005 Vegetation Cover Factor map(e), 2013 Vegetation Cover

Factor map(f)


**4   Result**
The above factor layers are converted into raster layers in 30 m×30 m of same coordinate under the
support of ArcGIS10.0 software. The layers are multiplied to obtain the spatial distribution of soil
erosion modulus in the study area. Soil erosion is graded by reference to SL190-2007 criteria for
classification of soil erosion intensity in the Classification of Soil Erosion, Ministry of Water



Resources(Fig.3).

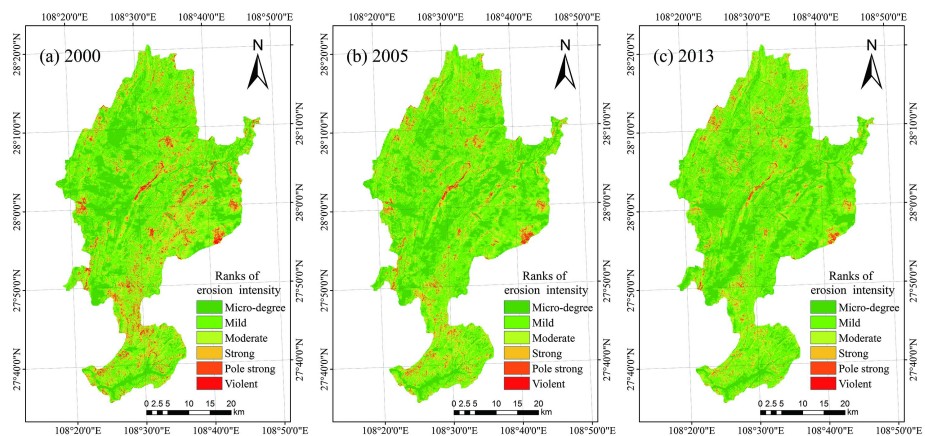

**Figure 3.** Spatial distribution of soil erosion in Yinjiang in different periods

## 4.1    Evolution of soil erosion in the study area

Result shows(Table 3) that in 13 years from 2000 to 2013, the total amount of soil erosion in Yinjiang was reduced from $477.48 \times 10^4$ t·a$^{-1}$ in 2000 to $366.56 \times 10^4$ t·a$^{-1}$ in 2005 and $314.64 \times 10^4$ t·a$^{-1}$ in 2013 respectively, with total reduction range of 34.11%.

**Table 3.** Conditions of soil erosion in Yinjiang in different periods

|  | Erosion rating | Erosion area(hm²) | Total soil loss($\times 10^4$ t) | Average modulus (t·hm$^{-2}$·a$^{-1}$) | Area ratio(%) | Erosion ratio(%) |
|---|---|---|---|---|---|---|
| 2000 | Micro-degree | 36187 | 8.47 | 2.30 | 28.97 | 1.77 |
|  | Mild | 87470 | 126.25 | 126 | 39.99 | 26.44 |
|  | Moderate | 40506 | 146.58 | 36.11 | 19.27 | 30.70 |
|  | Strong | 15719 | 98.88 | 62.88 | 7.78 | 20.71 |
|  | Pole strong | 7153 | 73.73 | 103.30 | 3.46 | 15.44 |
|  | Violent | 1244 | 23.57 | 184.80 | 0.54 | 4.94 |
| 2005 | Micro-degree | 56529 | 9.74 | 2.35 | 30.27 | 2.66 |
|  | Mild | 84898 | 117.30 | 13.92 | 43.90 | 32.00 |
|  | Moderate | 34362 | 120.91 | 35.23 | 17.76 | 32.99 |
|  | Strong | 10929 | 67.95 | 62.17 | 5.65 | 18.54 |
|  | Pole strong | 4352 | 44.67 | 102.70 | 2.25 | 12.19 |



|      |             |       |        |        |       |       |
|------|-------------|-------|--------|--------|-------|-------|
|      | Violent     | 338   | 5.99   | 177.59 | 0.17  | 1.64  |
| 2013 | Micro-degree | 63544 | 10.57  | 2.32   | 34.21 | 3.36  |
|      | Mild        | 85610 | 117.63 | 13.83  | 44.29 | 37.42 |
|      | Moderate    | 30801 | 107.54 | 34.97  | 15.92 | 34.21 |
|      | Strong      | 8010  | 49.73  | 62.11  | 4.14  | 15.82 |
|      | Pole strong | 2663  | 26.76  | 100.52 | 1.38  | 8.51  |
|      | Violent     | 125   | 2.11   | 168.55 | 0.065 | 0.67  |

For the soil erosion area, the area of micro erosion accounts for 28.97%, 30.27%, and 34.21%
of total erosion area in three study periods from 2000 to 2013, with a total increase of 5.24%. The
area of slight erosion accounts for 39.99%, 43.90%, and 44.29% of total erosion area respectively,
which was decreased by 1860 hm$^2$ in the study period but increased by 4.30% in percentage. The
sum of micro erosion are and slight erosion area reaches more than 65% in three periods, and the
percentage of moderate erosion and above shows a declining trend from 2000 to 2013. Among which,
the decreasing amplitude of moderate erosion area, strong erosion area, very strong erosion area, and
severely strong erosion area is 24%, 49%, 63%, and 89%, respectively. Yinjiang County exhibited a
transformation from moderate erosion, strong erosion, very strong erosion, severely strong erosion,
and above to micro and slight erosions.
For the soil erosion amount, the percentages of micro-erosion, slight-erosion, moderate-erosion
that amount to total erosion are increased during the study period. Slight erosion and moderate
erosion contribute to the erosion amount in Yinjiang County. The sum percentage of erosion is
increased from 57.14% in 2000 to 71.63% in 2013, and the percentages of strong erosion, very
strong erosion, and severely strong erosion are significantly decreased. The sum percentage of strong
erosion and very strong erosion is decreased from 36.15% to 24.33%.
In summary, the erosion amount in Yinjiang County is mainly concentrated in slight and
moderate erosions. The sum percentage of soil erosion amount from 2000 to 2013 is increased by
12.57%. In the whole Yinjiang County, a large scale of land undertook micro erosion and slight
erosion in 2000, 2005, and 2013. The sum of erosion scope is more than 65%. The corresponding
soil erosion amounts account for 28.21%, 34.66%, and 40.78% of the total erosion amount.
Although the total erosion area is increased to 2374 hm$^2$, the areas of micro erosion and above are



reduced. The erosion amount also shows a decreasing trend year by year, and the erosion level is
significantly changed from high to low in a large area.
**4.2    Grade shifting of soil erosion intensity in study area**
In 2000–2005, the percentage of the area with unchanged soil erosion intensity was 22.76%; the
percentage of the area with the increased soil erosion intensity was 33.68%; and the percentage of
total area with the decreased erosion intensity was 43.56%. This finding reveals that the soil erosion
level transformed from moderate and high levels to low level in this period.
In 2005–2013, the percentage of the area with unchanged soil erosion intensity was 23.19%,
which increased by 0.43% compared with that in 2000–2005. The percentage of the area with the
increased soil erosion intensity was 40.2%, and the percentage of the area with the decreased erosion
intensity was 36.59%. In addition, the percentages of the areas with increased and decreased erosion
intensity are slightly increased.

**Table 4.** The intensity variation of the soil erosion in the study area

| Grade shifting of soil erosion intensity(%) | | | | | | | | | |
| 0 | 1 | 2 | 3 | 4 | -1 | -2 | -3 | -4 | -5 |
|---|---|---|---|---|---|---|---|---|---|
| 2000-2005 | | | | | | | | | |
| 22.76 | 15.23 | 13.07 | 4.33 | 1.05 | 24.22 | 8.52 | 9.50 | 1.09 | 0.24 |
| 2005-2013 | | | | | | | | | |
| 23.19 | 17.77 | 21.15 | 1.02 | 0.26 | 13.93 | 14.28 | 6.19 | 2.11 | 0.08 |
| 2000-2013 | | | | | | | | | |
| 19.74 | 18.33 | 10.21 | 2.47 | 0.59 | 19.10 | 10.96 | 15.61 | 2.70 | 0.29 |

Note: 0 refers to the unchanged soil erosion intensity; 1 refers to the soil erosion intensity increased by one level; 2 refers to the soil erosion
intensity increased by two levels; 3 refers to the soil erosion intensity increased by three levels; 4 refers to the soil erosion intensity increased
by four levels; 5 refers to the soil erosion intensity increased by five levels; -1 refers to the soil erosion intensity decreased by one level; -2
refers to the soil erosion intensity decreased by two levels; -3 refers to the soil erosion intensity decreased by three levels; -4 refers to the soil
erosion intensity decreased by four levels; and -5 refers to the soil erosion intensity decreased by five levels.

In summary, the percentage of total area in 2000–2013 with increased erosion intensity was
31.6%, and that with decreased erosion intensity was 48.66%. This finding reveals that the soil
erosion intensity shows an improving trend.
**4.3    Spatial variation of soil erosion in the study area**
**4.3.1    Different slope zone**



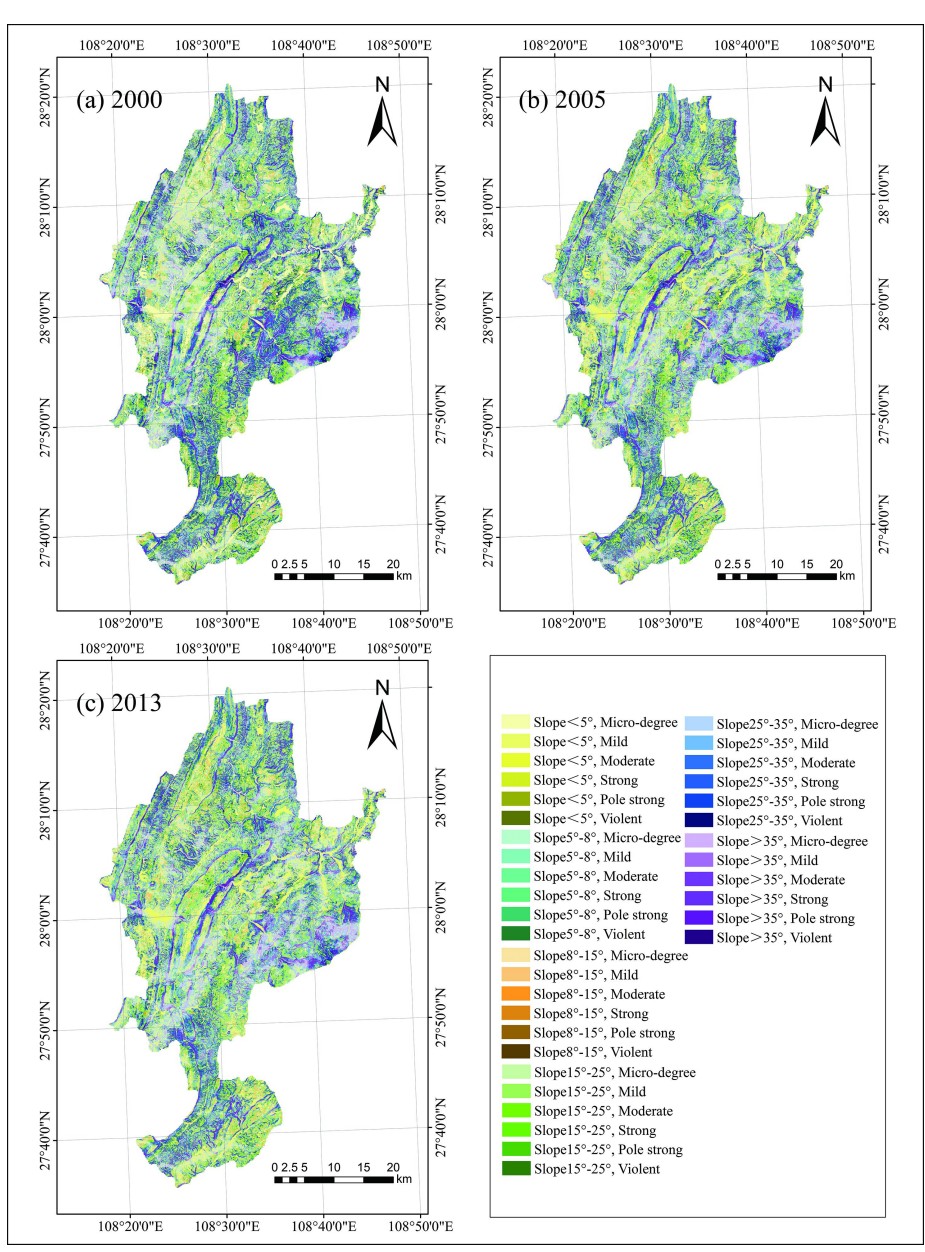

**Figere 4.** Spatial distribution of soil erosion in different slope band

Slope is the most important terrain factor that influences soil erosion. Soil erosion modulus is closely

related to slope. Soil erosion modulus in Yinjiang County gradually increases with the increase of



slope. This finding shows a significantly positive correlation. The mean soil erosion modulus in

high-slope area is higher, but the erosion area and erosion amount are smaller.

**Table 5.** Soil erosion conditions in different slope grades

| Slope | Average modulus($t \cdot hm^{-2} \cdot a^{-1}$) | Area ratio(%) | Erosion ratio(%) |
|---|---|---|---|
| <5° | 15.32 | 9.68 | 10.85 |
| 5°-8° | 13.31 | 4.76 | 17.32 |
| 8°-15° | 15.33 | 12.94 | 18.09 |
| 15°-25° | 17.56 | 33.31 | 19.68 |
| 25°-35° | 18.54 | 27.28 | 20.72 |
| >35° | 20.15 | 12.03 | 13.33 |

The soil erosion area is the largest in 15°–25° slope bands, accounting for 33.31%, followed by

25°–35° slope bands that account for 27.28%. For the percentage of erosion amount, 25°–35° slope

bands account for 20.71%, 15°–25° slope bands account for 19.68%, 8°–15° slope bands account for

18.09%, and 5°–8° slope bands account for 17.32%. The band with slope <5° has the lowest erosion

amount, accounting for 10.85%. For the mean erosion modulus, different slope bands are in

slight-erosion level.

### 4.3.2 Outcrop area of different rocks

The Karst surface is broken, with a great number of peak cluster, needle karst, and isolated peaks.

The area of carbonate rocks distributed in the study accounts for 60.06% of the total area. From 2000

to 2013, the annual erosion rate was reduced by 8.22 $t/(hm^2 \cdot a)$, with a decreasing amplitude of

30.82%. In non-carbonate rock areas, the annual erosion rate from 2000 to 2013 was reduced by 6.19

$t/(hm^2 \cdot a)$ with a decreasing amplitude of 24.29%, which is smaller than that in carbonate rock area.

For the carbonate rock area, the annual erosion rate during 13 years from 2000 to 2013 is as

follows: reduced by 12.24 $t/(hm^2 \cdot a)$ with a decreasing amplitude of 40.40% in the homogenous

dolomite(HD) area (allowable loss amount in the area T=20); reduced by 3.8 $t/(hm^2 \cdot a)$ with a

decreasing amplitude of 15.99% in the homogenous limestone(HL) area; reduced by 1.28 $t/(hm^2 \cdot a)$

with a decreasing amplitude of only 5.26% in the mixed area of homogenous limestone and

homogenous dolomite(MHLD); reduced by 4.38 $t/(hm^2 \cdot a)$ with a decreasing amplitude of 20.11% in

the clastic rock area of limestone interlayer(CRLI) (allowable loss amount in the area T=100); and



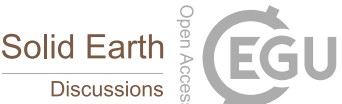

reduced by 4.31 t/(hm²·a) with a decreasing amplitude of 17.07% in the interbedded area of
limestone and clastic rock(ILCR) (allowable loss amount in the area T=250).

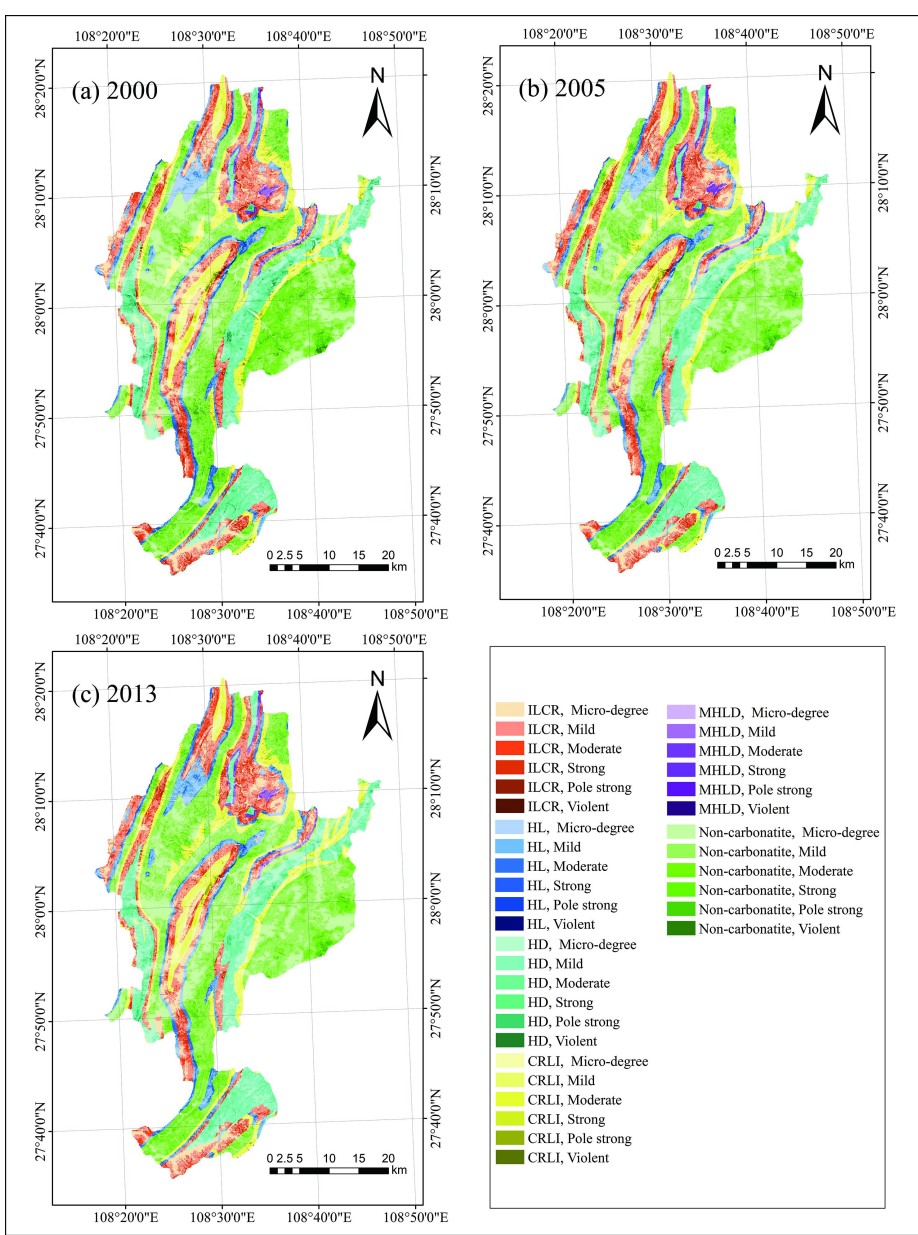

**Figure5.** Spatial distribution of soil erosion in different outcrop areas of rocks

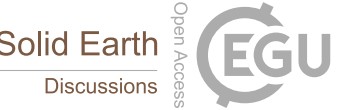

**Table 6.** Annual erosion rates in different outcrop areas of rocks

| | Average soil erosion rate(t·hm$^{-2}$·a$^{-1}$) | | | | | | |
|---|---|---|---|---|---|---|---|
| | Non-carbonatite | carbonatite | HD | HL | MHLD | CRLI | ILCR |
| 2000 | 26.67 | 25.48 | 30.30 | 23.77 | 24.34 | 21.78 | 25.25 |
| 2005 | 21.79 | 21.82 | 22.26 | 21.86 | 27.44 | 19.10 | 23.03 |
| 2013 | 18.45 | 19.29 | 18.06 | 19.97 | 23.06 | 17.40 | 20.94 |

For the change in decreasing amplitude in the study period, the relationship is as follows: continuous dolomite (T=20) > clastic rock of limestone interlayer (T=100) > interbedded of limestone and clastic rock (T=250) > homogenous limestone > mixture of homogenous limestone and dolomite.

### 4.3.3 Different rocky desertification grades

Different degrees of rocky desertification are distributed in about 57.69% of the study area. Under the background of Karst, the interference and destruction of unreasonable social and economic activities caused severe soil erosion, which leads to soil particle loss in desertification area, thinner soil layer, and outcropped base rock.

**Table 7.** Annual erosion rate in different rocky desertification grades

| | Average soil erosion rate(t·hm$^{-2}$·a$^{-1}$) | | | | | |
|---|---|---|---|---|---|---|
| | None RD | Micro RD | Mild RD | Moderate RD | Severe RD | Non-karst |
| 2000 | 30.46 | 25.40 | 21.48 | 18.54 | 9.71 | 25.93 |
| 2005 | 22.17 | 21.79 | 20.09 | 18.57 | 8.98 | 21.74 |
| 2013 | 18.47 | 19.17 | 18.28 | 16.86 | 11.56 | 18.51 |

In 2000–2013, the annual erosion rate in Yinjiang County was as follows: reduced by 11.99 t/(hm$^2$·a) with a decreasing amplitude of 39.36% for the non-rocky desertification area; reduced by 6.23 t/(hm$^2$·a) with a decreasing amplitude of 24.53% for the micro rocky desertification area; reduced by 3.2 t/(hm$^2$·a) with a decreasing amplitude of 14.90% for the slight rocky desertification area; reduced by 1.68 t/(hm$^2$·a) with a decreasing amplitude of 9.06% for the moderate rocky desertification area; increased by 1.86 t/(hm$^2$·a) with an increasing amplitude of 19.16% for the severe rocky desertification area; and reduced by 7.42 t/(hm$^2$·a) with a decreasing amplitude of 28.62% for the non-rocky desertification area.



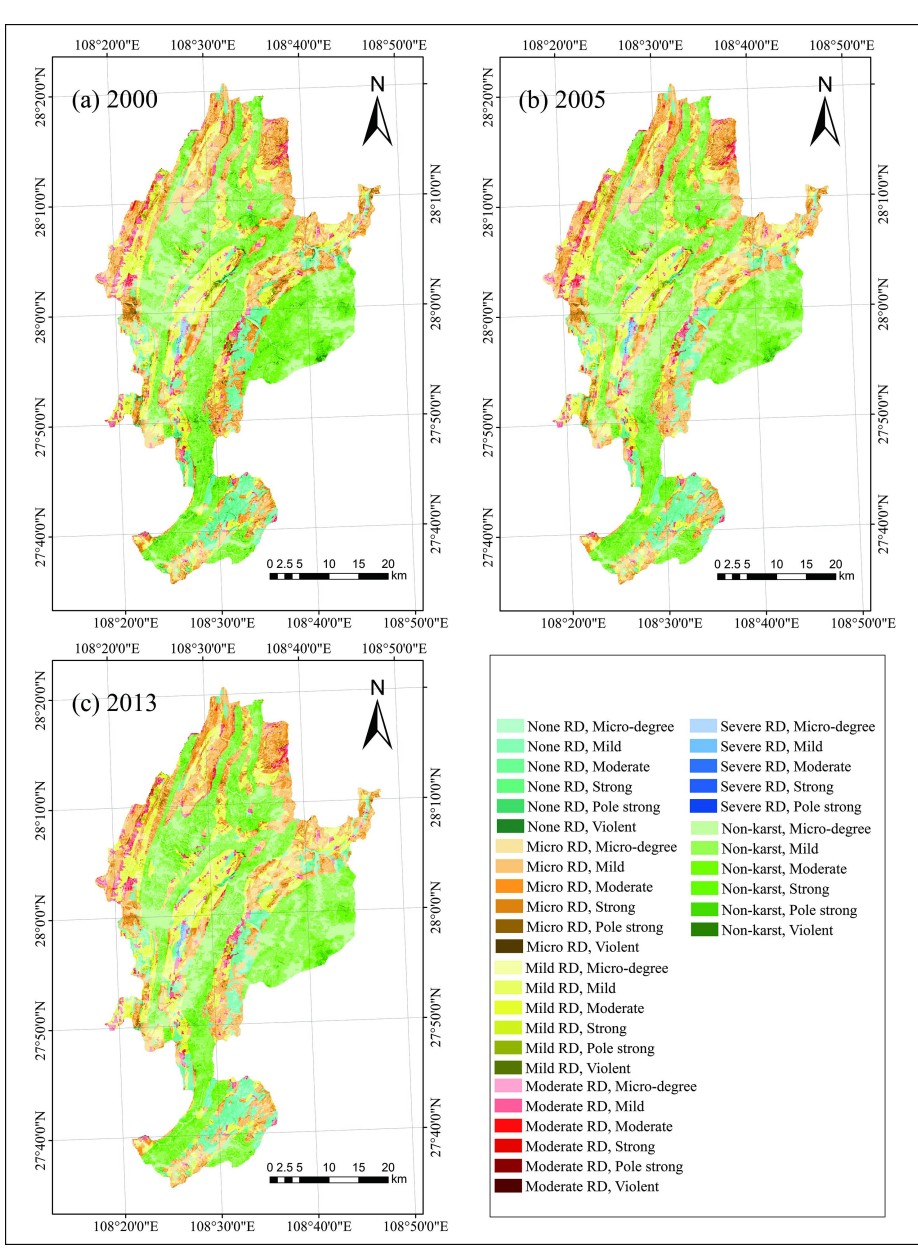

**Figure 6.** Spatial distribution of soil erosion in different rocky desertification grades

The relationship of the decreasing amplitude of erosion rate of Karst areas in the study period is as follows: non rocky desertification area > micro rocky desertification area > slight rocky

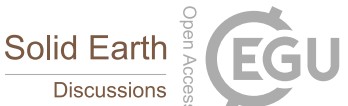

desertification area > moderate rocky desertification area > severe rocky desertification area. During
the study period, the soil erosion amounts in non-rocky desertification area, micro rocky
desertification, slight rocky desertification, and moderate rocky desertification area showed a
declining trend, whereas that in severe rocky desertification showed an increasing trend. In the study
area, micro erosion occupied the largest soil erosion area (47.55% of the total area) and has the
highest erosion amount (48.86% of the total erosion amount). The mean erosion modulus is in the
level of slight erosion.

**5  Discussion**
**5.1  Spatio temporal evolution characteristics of soil erosion**
The overall soil erosion condition in Yinjiang County was yearly improved. The erosion area and
erosion amount are represented by the conversion from strong, very strong, and severely strong
erosion to moderate erosion and below. This phenomenon occurs because rainfall and vegetation
coverage are the major factors that affect the dynamic changes of soil erosion in Yinjiang County.
On the one hand, rainfall was yearly reduced during the study period, from 1121.03 mm in 2000 to
734.39 mm in 2013; hence, the rainfall erosion was weakened. On the other hand, Yinjiang County
has a wide range of returning farmland to forests, and the closed forest project, so vegetation
management and soil- and- water conservation measures in the study area are correspondingly
changed. The vegetation coverage is improved and thus plays a role in the prevention and control of
soil-and-water erosion. Soil-and-water measures have active effects and cause significant results.
Different slopes determine different speeds of surface runoff. If other factors are unchanged, in
the area with the slope below 35°, with the increase of slope, the washing of surface runoff on soil
become stronger, so as to increase soil erosion amount. When the slope is up to 35°, erosion amount
shows a declining trend, weakly influenced by the increasing slope. The band with the slope of
15°–35° accounts for 60.59% of the total erosion area and 40.44% of the total erosion amount. This
band is the main erosion slope section in the study area. This phenomenon is the result of artificial
reclamation in the slope area. Combined with previous studies(Xu et al., 2008; Chen, 2012), this



slope area in in Yinjiang County must have enhanced prevention and control measures for soil
erosion.

### 5.2  Influence of spatial factors on soil erosion

#### 5.2.1  Influence of lithology on soil erosion

The decreasing amplitude of soil erosion rate in carbonate area is larger than that in non-carbonate
area. This finding is related to the widely distributed rocky desertification in the Karst area, the soil
forming rate, the soil types, and other factors. After the carbonate rock is dissolved in the study area,
the soluble matter is removed by water and the insoluble matter forms the soil. The content of
insoluble matter in carbonate rock in the southwest area is 1%–9%, generally less than 5%. The soil
forming efficiency is low. After erosion and weathering, 630–7880 ka of carbonate is required to
form 1m thickness soil layer. The soil forming rate is 10–40 times slower than that in general
non-Karst area(Chen, 1997). The soil forming rate and soil thickness are higher in non-carbonate
area than those in carbonate area. The formation time of runoff is short after rainfall and the surface
water storage capacity is poor in Karst area. Much rainfall is formed in the underground runoff;
hence, the underground soil loss is high and the vegetation coverage is lower than that in non-Karst
area.
In the study period, only 10%–22.37% of the areas are within the allowable loss amount. These
areas are mainly distributed in the valley zone with lower altitude in the south of Yinjiang, and the
smooth zone in southwest area and Fanjingshan area. These areas are mostly located in non-Karst
area with widely distributed non-carbonate. The soil forming rate is rapid. The underground soil loss
is low and the vegetation coverage is high.
The soil erosion in different outcrop areas generally shows an improving trend. However, the
dynamic change in soil erosion in various lithological distribution belts is significant. The
decreasing amplitude of the annual erosion rate in homogenous dolomite, limestone intercalated
with clastic rock, interbedded region of limestone, and clastic rock is gradually reduced with the
decreasing content of carbonate. This phenomenon occurs because the mineral composition and





chemical characteristics of the parent rock directly affect the speed and direction of soil formation.
The weathering degree of different lithologies, the speed and direction of soil forming process, and
the erosion way, erosion intensity, and rate are also different. If the content of the carbonate is
higher, then the soil forming rate is slower and the soil layer is shallower. Therefore, the decreasing
amplitude of annual erosion rate is smaller. The homogenous limestone region and the mixed
region of homogenous dolomite and limestone are mainly distributed in the area in of low altitude
with slope less than 8°. Therefore, a certain soil thickness exists, which results in larger erosion
model and smaller decreasing amplitude of annual erosion rate. Moreover, the lithology also
controls the spatial distribution and development of soil erosion. The study of Li Yangbin et
al.(2006) shows that the allowable soil loss is 6.75 t/(km$^2$·a) in carbonate area and 7.08 t/(km$^2$·a) in
homogenous limestone area and homogenous dolomite area, and the rank of allowable loss
amounts is as follows: homogenous dolomite composition distribution area > homogenous
limestone composition distribution area. The rank of calculated loss amounts (homogenous
dolomite area > homogenous limestone area) in the current study is consistent with the previous
study. The allowable soil loss amount in limestone intercalated with clastic rock is 45.40 t/(km$^2$ ·a),
whereas that in interbedded region of limestone and clastic rock is 103. 38 t/(km$^2$ ·a). The
relationship of the allowable loss amount is: interbedded region of limestone and clastic rock >
limestone intercalated with clastic rock, which is positively correlated to the loss amount calculated
in areas of T=100 (limestone intercalated with clastic rock) and T=250 (interbedded layer of
limestone and clastic rock).

### 5.2.2   Effects of rocky desertification on soil erosion

In terms of soil erosion intensity in the study area, the decreasing amplitude of annual soil erosion
rate is gradually reduced with the aggravation of rocky desertification. When the degree of rocky
desertification is higher, the erosion modulus is lower and the decreasing amplitude of annual
erosion rate is smaller. The decreasing amplitude of annual erosion rate in non-rocky desertification
area is higher than that in rocky desertification area. This phenomenon occurs because the non-rocky
desertification areas are mainly distributed in valley and low-altitude regions with a certain thickness



of soil and good vegetation coverage. At present, the soil erosion rate in severe rocky desertification
in the study is increased with insignificant large loss intensity (total amount of soil erosion is small
and low). This phenomenon occurs because these areas are concentrated in Langxi valley, a small
distributed area with poor conditions of growing vegetation, or these areas are a negative relief in the
soil handling accumulation environment. The certain soil thickness causes the high erosion rate.

Erosion rates in other rocky desertification bands are reduced. This finding reveals that the soil

erosion in the rocky desertification area improved during the study period. The reason for soil loss
in the Karst rocky desertification areas are the particular geological (wide distribution of carbonate
rocks), topographical (the existence of underground space), vegetation, and climate conditions that
lead to low soil forming rate and shallow soil layer in the study area. Abundant rainfall in the study
area provides the dynamic potential for soil and water loss. However, underground pores, cracks,
and pipes are widely distributed in the Karst area. In addition to surface loss, soil loss also occurs in
Karst cave, underground rivers, and other ways. Therefore, the current study method has a certain
limitation in typical Karst area. In future studies, the underground soil loss should be calculated.
The localization of model calculation factor in Karst area should be considered in calculating the
soil erosion in Karst areas by using the model. The method improvement of the particularity of soil
erosion in the Karst area and the exploration of erosion indicators are performed to improve and
enrich the study on soil erosion in Karst area.

## 6   Conclusions

The temporal and spatial variations of soil erosion in the study area are gradually declining. These
variations show a changing trend from moderate level and above to the below level. Slope is the
most important topographic factor that causes different spatial and temporal distributions of soil
erosion. The band with the slope of 15°–35° is the main erosion slope section in the study area. The
soil erosion in rock outcrop area shows an improving trend, but the dynamic change in soil erosion
in each lithological distribution zone greatly varies. If the rocky desertification degree is higher, then
the erosion modulus is lower and the decreasing amplitude of annual erosion rate is smaller.

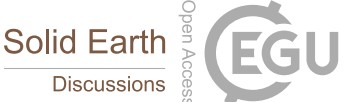

In Karst areas, the lithology and rocky desertification are the most important natural factors that
cause different temporal and spatial variations of soil erosion. Lithology is the geological basis of
soil erosion, and rocky desertification is widely distributed in Karst valley area. Different spatial
distributions of lithology and rocky desertification lead to a large area of soil loss. Lithological and
rocky desertification factors introduced in soil erosion model accurately reflect and predict the soil
erosion conditions and spatial distribution characteristics in Karst areas. This finding will help
promote the research on soil erosion in global Karst areas.
In Karst areas, underground space is developed. In addition to surface loss, soil loss is also
occurs in Karst cave, underground rivers, and other ways, causing the differences between the
measured soil loss and the calculated value by the model. Most of the time, the soil erosion study
method and indicators used for non-Karst area cannot reflect the actual situations of the Karst area.

**Acknowledgements.** This research work was supported jointly by National Key Research Program
of China (No. 2016YFC0502300, 2016YFC0502102, 2013CB956700 & 2014BAB03B02 ),
International cooperation research projects of the national natural science fund committee
(No. 41571130074 & 41571130042) , Science and Technology Plan of Guizhou Province of China
(No. 2012-6015),Agricultural Science and Technology Key Project of Guizhou Province of  China
(No. 2014-3039), Science and technology cooperation projects(No. 2014-3), Science and
Technology Plan Projects of Guiyang Municipal Bureau of Science and Technology of China
(No. 2012-205).

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
