# Peer review of "Soil erosion evolution and spatial correlation analysis in a"

_Solid Earth, 2017_

## Referee Comment (RC1) · Anonymous Referee #1 · 26 Jan 2017

This is a carefully done study and the findings are of considerable interest. In this paper, the authors presented some interesting results on the soil erosion evolution and spatial elements of soil erosion in a typical karst geomorphology, and possible driving forces behind the observed changes in soil erosion between 2000 and 2013. The authors attempted to present some qualitative explanations on the spatial impact of soil erosion from detected historical rocky desertification and lithology elements change in Yinjiang County, and they attributed Changes in soil erosion to both impacts of rocky desertifications and lithology. There has been much work done related to soil erosion over area across the world. But studies conducted on a typical karst area is currently lacking. From this regard, the manuscript is interesting and useful. For the benefit of

the reader, however, a number of points need to be clarified and certain statements require further justification, as shown below: 1) The language in s articles must be clear, correct, and unambiguous. Any typographical or grammatical errors should be corrected at revision, so please note any specific errors here. 2) The text must be carefully checked by the authors as it contains several errors. 3) Lines 85-98 References in this part of research background must be updated to nearly five-year research, in order to stand out the forefront of your study. 4) In the part of Study area, it will be better if you can split Figure.1 into two pictures(Location and geological background), which can more clearly express the thinking of your work. 5) In the part of 4.2(Grade shifting of soil erosion intensity in study are), Table.4 is replaced by figure will make your paper better. 6) In the part of DiscussionïijŇit is recommended to add some documents to prove that your research is reasonable. 7) Although this paper is good, it would be ever better if some comparison and validation of existing research results were added. In view of the aforementioned issues, this manuscript can be acceptable after revisions as recommended above.

---

## Referee Comment (RC2) · Anonymous Referee #2 · 27 Jan 2017

OVERALL COMMENT: 1. This manuscript presents a mere application of RUSLE to interpret the variations of soil erosion in a Karst area. I do think that is a good try to improve the management strategies in the region, however, I cannot recommend its publications due to: i) its scientific interest may be debatable because there is no evidence to prove their findings; it is definitely a very general study; ii) In addition, the formal aspect presents clearly room for improvement , namely objectives, material&method and results were not well-linked. I also recommend the English language revision.

DETAILED COMMENTS: 2. Abstract: -Line 19, where you evaluated the correlation degree? What type of spatial elements? -Line 25 and 62, what is exactly rocky desertification? - Please, I do think that the bullet points are appropriate in the abstracts. -Line 29, 15-35° are equivalent to 25 -70 % which is a very steep slope range. I do think the explanation is evident. - Lines 36-37, lithology is evidently the geological basis of the erosion. . .

3. Introduction: - Lines 49-52, please review the text, you don't explain the problem to solve, only mention that is very complex. - Line 54, why plots are not useful, at least, they were useful to provide actual measurements. - Line 72, why fragile? - Lines 86-93, I recommend to be respectful with the publications of other authors and you integrate your work into the knowledge chain. Why deficiencies? Miscarriage of justice? I'm afraid this must be a very bad translation. - Line 101, what type of "actual survey" did you carry out? - Line 105, where you study the ecological effects?

4. Material and Methods -Please, it is essential to link data and methods to achieve your objectives. They were not connected. You have not explained your steps to reach your results and in addition, material and methods were mixed. For instance, lines 226-229 must be included into M&M. -The study region is of 1969 ha, however, you don't discuss the applicability of the equations you used. For instance, as for erosivity, how did you calculate the spatial average rainfall? How many gauges did you consider? Is there a sharp variability? And the erodibiliy? What type of data were used? It is particularly important to interpret your results. - The length of dataseries is not clear, why in Table 1 is there only 3 years. More information about the dataseries should be included to interpret the analyses.

5. Results, discussion and conclusions - I would like to encourage the authors to present another more specific work about the features of the most fragile areas and their spatial complexity. I see you have a very high annual precipitation (1100 mm), a steep topography which both imply a great deal of available energy combined with a high erodibility (soft materials). Therefore, if you don't have vegetation or protection to dissipate so much energy on the soil, risk of soil losses are evidently clear. Of course, if the soil materials were more resistant, the expected soil losses would be lower. On the

other hand, rocks must contribute to protect soil as mulch or because they constitute a tough material in outcrops, however, these aspects were not well-developed. It must not be straightforward at your work scale. - Anyway, the chapters 4.3.3. and 5 are very difficult to follow with the tables and figures presented. Rocky desertification and the history of land uses should have been explained into the introduction.
* * *

---

## Referee Comment (RC3) · Anonymous Referee #3 · 30 Jan 2017

The authors present a classical RUSLE application in a Karst area in China. The paper gives a nice overview of soil erosion estimation in karst regions.

Although, the use of the RUSLE is disputed in soil erosion community because of its limitations and weaknesses, there is still no better alternative for modelling the estimation of long-term soil erosion on disturbed hillslopes.

It is an interesting study, clear, concise and within the scope of the journal. I only have a comment to improve discussion and recommend this manuscript for publication in Solid Earth after minor revisions.

The introduction is well written including previous scientific work, the relevance of topic

and specific objectives. Also study area as well as material and methods are well described.

Strongly required is a better discussion with a comparison to other studies using RUSLE. Please discuss your findings with current knowledge of soil erosion in karst regions and in general. In the current form it is more an interpretation than a discussion!!!

Please improve the language. There are some careless mistakes throughout the manuscript.

I did not check the correctness of the equations. Authors refer to existing literature, therefore they should be correct.

I have not checked in detail if authors adequately adhere to the journal's Guide for Authors.

I have not checked plagiarism.

―――――――――――――――――――――――

---

## Referee Comment (RC4) · Anonymous Referee #4 · 4 Feb 2017

Overall

This paper used the GIS technology and RUSLE model to analyze the spatial and temporal evolution characteristics of soil erosion and investigate the relationship between soil erosion and rocky desertification in Karst region. The soil erosion in Karst region is very different from other soil erosion in that its unique land surface characteristics. It is an interesting work for understanding the soil erosion in this region so I strongly recommend it to be published in this journal. However, my general opinion is that this paper still needs further improvements. The main shortcoming of this paper is that the descriptions are somewhat unclear. I suggest some major revisions should be made in paper descriptions before publication.

[Figure]

Comments (1) In Abstract, the background description should be concise; (2) In Introduction, the status of current research should describe clearer; (3) In RUSLE model should described more concise; (4) It is better in the Results section to clearly describe results from data analysis or from model simulation; (5) In Discussion section, the contribution of this paper should be emphasized in first paragraph. And some uncertainties should be addressed. (6) The English should further smooth.

---

## Author Comment (AC1) · 10 Feb 2017

Dear reviewer: I am very grateful to your comments for the manuscript. According with your advice, we amended the relevant part in manuscript. Some of your questions were answered below. In this manuscript, the influence of spatial factors was studied which included the topography, lithology, rocky desertification, and et al. on soil erosion in typical Karst area, Because of the special geological background, the study of Karst area is relatively small, this paper is an interesting research. By comparing the results with other RUSLE models, this paper illustrated the particularity and necessity of this research work, and also verified the correctness of the work. I have also made a careful revision of the expressions and language issues you have raised.

Many grammatical or typographical errors have been revised. Thank you and all the reviewers for the kind advice. Sincerely yours, Cheng Zeng Corresponding author: Name: Xiaoyong Bai E-mail: baixiaoyong@126.com

Please also note the supplement to this comment:
http://www.solid-earth-discuss.net/se-2017-1/se-2017-1-AC1-supplement.pdf

---

## Author Comment (AC2) · 10 Feb 2017

Dear reviewer: I am very grateful to your comments for the manuscript. According with your advice, we amended the relevant part in manuscript. Some of your questions were answered below. 1)The abstract will be described its main points which included the ăbackground (the necessity and ăimportance in strengthen the research on soil erosion in karst region), objectives, methods, results, and conclusion. 2)The part of the introduction At the beginning, it explains the significance of the research ; Previous studies including the fact that remote sensing is used to obtain the driving mechanism; However, there is no clear understanding of soil erosion and spatial factors, it is very important to understand the characteristics of soil erosion in Karst. In this article, we

try to make it clear by means of the RUSLE model and the GIS . 3)We can simplify the description of the RULSE model, the description is divided into two main parts: a) what is the model; b)Model RULSE calculation process 4)In the part of the results, firstly the results of soil erosion are analyzed, then the factors such as topography, lithology and rocky desertification were analyzed, Study on the relationship between soil erosion and spatial and temporal factors. Thank you for your adviceïijŇI will clearly describe results from data analysis and from model simulation; 5)In the discussion section, the characteristics of soil erosion and its relationship with spatial elements (lithology, rocky desertification) are discussed in several parts, the causes of the problem are discussed in each section, and the contributions are summarized. 6)Many grammatical or typographical errors have been revised. All the lines and pages indicated above are in the revised manuscript. Thank you and all the reviewers for the kind advice. Sincerely yours, Cheng Zeng Corresponding author: Name: Xiaoyong Bai E-mail: baixiaoyong@126.com

---

## Author Comment (AC3) · 11 Feb 2017

Dear reviewer: I am very grateful to your comments for the manuscript. According with your advice, we amended the relevant part in manuscript. Some of your questions were answered below. 1)Thank you for your careful examination of the article, I have the article in the language of many adjustments and modifications to make the expression more clear, correct and unambiguous. 2)The spelling and syntax errors have been checked and corrected. 3)In line 85-98, the literature on soil erosion research in the nearly five years has been added, and analyzed and summarized in this paper. 4)Figure 1 shows position and geological conditions of the study area, it is divided into two picture can indeed make this part of the expression is more clear, so

following the reviewer's comment , the Figure 1 divided into two pieces, when the final layout editing needs which. 5)Table.4 is replaced by figure will make the manuscript better and more intuitive, some people have done this before. 6)In the discussion part, the missing reference has been added into the revised manuscript. 7)Following the reviewer's comment, a new sub-section(5.3) has been added to the manuscript. to address the statistical analysis for the use of RUSLE model of Karst soil erosion research. Thank you for the kind advice. Sincerely yours, Cheng Zeng Corresponding author: Name: Xiaoyong Bai E-mail: baixiaoyong@126.com

Please also note the supplement to this comment:
http://www.solid-earth-discuss.net/se-2017-1/se-2017-1-AC3-supplement.pdf

**Supplement:**

**5.3 Modulus of Different Soil Erosion Statistics in Karst**

AreasThe RUSLE model is one of the classical models of soil erosion and is widely used in various countries and regions in the world. However, although the RUSLE model is a mature and classical model, its application in karst areas is relatively scarce. Several scientists have conducted research on different parts of the karst areas of Guizhou Province. Different results have been derived; thus, a simple control should be adopted, and their results are given in Table 8.

**Table 8 .**The different data of soil erosion obtained by previous studiesin typical Karst area

| The main author | Study area | Time scale | Average modulus ($t \cdot hm^{-2} \cdot yr^{-1}$) | Total soil loss ($\times 10^4$ t) |
|---|---|---|---|---|
| Zeng(2011) | Hongfenghu Basin | 1960-1986 | 38.35 | 610.53 |
| | | 1987-1997 | 52.80 | 839.90 |
| | | 1998-2004 | 40.24 | 640.18 |
| Xu(2011) | Maotiao River Basin | 2002 | 28.70 | 875.65 |
| Wang(2014) | Wujiang River Basin | 1980-1989 | 26.78 | 133.36 |
| | | 1990-1999 | 23.13 | 115.18 |
| This paper | Yinjiang Country | 2000 | 25.09 | 477.49 |
| | | 2005 | 21.53 | 366.56 |
| | | 2013 | 18.84 | 314.64 |

---

## Author Comment (AC4) · 21 Feb 2017

Dear Reviewers: Thank you for your concerning our manuscript. Those comments are all valuable and very helpful for revising and improving our paper, as well as the important guiding significance to our researches. Some of your questions were answered below.

1.OVERALL COMMENT: i)The most important environmental problem in Karst is the loss of soil and water, which causes serious social, environmental and economic problems. Although the predecessors have done a lot of work in this area, but there are soil erosion and topography, lithology, rocky desertification and other spatial factors are not clear, clear this is very important for soil erosion control in Karst. This article tries to

[Figure]

make it clear by means of RUSLE model and GIS. Following the reviewer's comment, a new sub-section(5.3) has been added to the manuscript, in order to verify our work; ii)We will carefully sort out these parts to make their relationship more reasonable. And the spelling and syntax errors have been checked and corrected.

2.DETAILED COMMENTS: Abstract: -Line 19, in this manuscript, the soil erosion in typical Karst area is calculated, and the characteristics of spatial variation and the correlation analysis of spatial elements are analyzed. The spatial elements are mainly related to the soil erosion, including topography, lithology, rocky desertification and so on. -Line 25 and 62, it is necessary to increase the interpretation of rocky desertification. Rocky desertification refers to the fragile ecological environment in Karst, as a result of the social economy unreasonable human activities caused by the obvious contradiction between people and land, vegetation destruction, soil erosion, decline or loss of the production capacity of the land surface, showing evolution of desert like landscape rocks gradually exposed. -Line29, the results of 15-35 degrees are obtained by calculation. The slope of 15-35 degree is a region affected by human activities, which is consistent with the results of previous studies (the area around 25 degrees is the area of soil erosion prone). - Lines 36-37, the lithology determines the formation of soil parent material, soil erosion in different soil types are different, so the lithology is the geological foundation soil erosion produced.

3.Introduction: - Lines 49-52, In this paper, the spatial variation of soil erosion in Karst area is studied. It is also because of the special natural environment in Karst area, which leads to the fact that the soil erosion is more special than other regions. - Line 54, we are very sorry for our negligence. the meaning of the original sentence is consistent with that of the reviewer,These methods are useful, but are difficult to apply to the Karst area.These methods can also provide references for this article. - Line 72, because the carbonate rocks in Karst area are widely distributed, the slow process of soil formation, thin soil, is unfavorable for the growth of vegetation, the ecological environment destruction once it is very difficult to recover, so the ecological environment

of Karst area is very fragile. - Lines 86-93, thank you for your advice, we will analyze and summarize the results of other authors, and strive for better integration with the research in this paper. Most of the previous studies in the Karst river basin or mountain area; Few scholars have studied the response of rock desertification and lithology to soil erosion, at the same time, few scholars have applied the influence of spatial factors on the evolution of soil erosion in Karst area. - Line 101, during the thesis writing and writing before, we conducted a number of field investigation in Yinjiang County, which relates to the meteorological characteristics of Yinjiang County, the lithology, rock desertification, topography, soil type, vegetation type. - Line 105, this paper studied the characteristics of soil erosion in Yinjiang county and its spatial factors on soil erosion, soil erosion will have an impact on the ecological environment, the ecological environment will also respond to soil erosion characteristics, including the impact of human activities on soil erosion, soil erosion effect on vegetation etc..

4.Material and Methods Thank you this proposal, The description of the data sources and the description of the method, we will follow your recommendations to improve. And the source of the data corresponds to the steps of the calculation, so that each part of the materials and methods have a very good correspondence. lines 226-229 part of the material can be described in the M&M section after a brief description of the source. The study region is of 1969 ha, the rainfall erosivity in the study area is calculated by using Zhou Fujian's formula based on monthly rainfall. The values of 2000, 2005 and 2013 were calculated respectively, and the average monthly precipitation was calculated. This equation is applicable to the case of Rainfall Erosivity in southern china.

5.Results, discussion and conclusions The study area in southern China, there are very abundant precipitation, however, the widespread distribution of Karst landform, precipitation will be quickly transferred to the underground through surface cracks, resulting in the precipitation of Karst is difficult to use. And the chapters 4.3.3. and 5ïïjŇthe soil forming rate of Karst area is small, the soil layer is thin, and the

soil erosion is very small. Therefore, there is no obvious difference between the different soil surface erosion rates controlled by lithology. This is because there is a large number of underground leakage of soil in Karst. We would like to express our great appreciation to you for comments on our paper. Thank you and best regards. Sincerely yours, Cheng Zeng Corresponding author: Name: Xiaoyong Bai E-mail: baixiaoyong@126.com

Please also note the supplement to this comment:
http://www.solid-earth-discuss.net/se-2017-1/se-2017-1-AC4-supplement.pdf

[Figure]

**Supplement:**

**5.3 Modulus of Different Soil Erosion Statistics in Karst**

AreasThe RUSLE model is one of the classical models of soil erosion and is widely used in various countries and regions in the world. However, although the RUSLE model is a mature and classical model, its application in karst areas is relatively scarce. Several scientists have conducted research on different parts of the karst areas of Guizhou Province. Different results have been derived; thus, a simple control should be adopted, and their results are given in Table 8.

**Table 8 .**The different data of soil erosion obtained by previous studiesin typical Karst area

| The main author | Study area | Time scale | Average modulus ($t \cdot hm^{-2} \cdot yr^{-1}$) | Total soil loss ($\times 10^4$ t) |
|---|---|---|---|---|
| Zeng(2011) | Hongfenghu Basin | 1960-1986 | 38.35 | 610.53 |
| | | 1987-1997 | 52.80 | 839.90 |
| | | 1998-2004 | 40.24 | 640.18 |
| Xu(2011) | Maotiao River Basin | 2002 | 28.70 | 875.65 |
| Wang(2014) | Wujiang River Basin | 1980-1989 | 26.78 | 133.36 |
| | | 1990-1999 | 23.13 | 115.18 |
| This paper | Yinjiang Country | 2000 | 25.09 | 477.49 |
| | | 2005 | 21.53 | 366.56 |
| | | 2013 | 18.84 | 314.64 |